# A Role for Gut Microbiome Fermentative Pathways in Fatty Liver Disease Progression

**DOI:** 10.3390/jcm9051369

**Published:** 2020-05-07

**Authors:** Paula Iruzubieta, Juan M. Medina, Raúl Fernández-López, Javier Crespo, Fernando de la Cruz

**Affiliations:** 1Gastroenterology and Hepatology Department, Marqués de Valdecilla University Hospital, Clinical and Translational Digestive Research Group, IDIVAL, 39008 Santander, Spain; p.iruzubieta@gmail.com; 2Instituto de Biomedicina y Biotecnología de Cantabria (IBBTEC), Universidad de Cantabria, 39011 Santander, Spain; jmedina@idival.org (J.M.M.); raul.fernandez@unican.es (R.F.-L.); delacruz@unican.es (F.d.l.C.)

**Keywords:** NAFLD, gut microbiome, microbial metabolic pathway, microbiome-based signature, fecal microbiota transplantation

## Abstract

Non-alcoholic fatty liver disease (NAFLD) is a multifactorial disease in which environmental and genetic factors are involved. Although the molecular mechanisms involved in NAFLD onset and progression are not completely understood, the gut microbiome (GM) is thought to play a key role in the process, influencing multiple physiological functions. GM alterations in diversity and composition directly impact disease states with an inflammatory course, such as non-alcoholic steatohepatitis (NASH). However, how the GM influences liver disease susceptibility is largely unknown. Similarly, the impact of strategies targeting the GM for the treatment of NASH remains to be evaluated. This review provides a broad insight into the role of gut microbiota in NASH pathogenesis, as a diagnostic tool, and as a therapeutic target in this liver disease. We highlight the idea that the balance in metabolic fermentations can be key in maintaining liver homeostasis. We propose that an overabundance of alcohol-fermentation pathways in the GM may outcompete healthier, acid-producing members of the microbiota. In this way, GM ecology may precipitate a self-sustaining vicious cycle, boosting liver disease progression.

## 1. Introduction

Non-alcoholic fatty liver disease (NAFLD) is the most common liver disease worldwide. Its prevalence is estimated at between 25% and 40% of adults, and its incidence is growing, probably as a result of the increase in obesity and associated metabolic disorders [1]. NAFLD includes a spectrum of pathological situations, ranging from simple steatosis to non-alcoholic steatohepatitis (NASH), fibrosis and cirrhosis. The later conditions potentially lead to hepatocellular carcinoma (HCC) [2]. The reasons why only a fraction of patients progress from NAFLD to NASH and cirrhosis are not understood, but obesity and insulin resistance are thought to be involved [3,4]. NAFLD is a multifactorial disease in which environmental, genetic, metabolic and inflammatory factors are involved. Among them, the gut microbiome (GM) is believed to be a key player, yet the pathogenic mechanisms involved are not entirely understood [5,6,7]. A recent theory on the pathogenesis of NAFLD postulates the involvement of “multiple parallel hits”. This hypothesis suggests that molecular mediators from various organs, particularly the adipose tissue and the gut, participate in triggering inflammation pathways, which may later progress to fibrosis and, eventually, carcinogenesis [8]. 

In recent years, increasing attention has been paid to the role of the GM in NAFLD pathogenesis. The GM comprises all microorganisms (bacteria, viruses, archaea, unicellular and pluricellular eukaryotes) in the digestive tract, forming a complex ecosystem that establishes a “symbiotic whole” with its human host. The GM plays a fundamental role in multiple physiological processes, including energy metabolism and immunological functions [9,10]. The human GM is dominated by four bacterial phyla: Bacteroidetes, Firmicutes, Proteobacteria and Actinobacteria. Of these, Bacteroidetes and Firmicutes are the most abundant [11]. Alterations in the microbiome composition have been associated with the development of chronic metabolic conditions, such as type 2 diabetes, obesity and NAFLD [7,12,13,14]. The influence of diet on GM composition and function is well established, and the GM seems to have the potential capacity to affect metabolic regulation of glucose and lipids in the host, through pathophysiological factors that may contribute to the development of metabolic syndrome and NAFLD [15].

## 2. The Gut Microbiome in NAFLD

Accumulated evidence indicates that the GM interacts with the liver via the so-called the “liver–gut axis” [16,17,18]. Dysfunction of this axis, including gut microbial imbalances and mucosa permeability alterations, leads to the passage of metabolic byproducts of bacterial metabolism as well as microbial components to the portal system reaching the liver. The microbial components, called pathogen-associated molecular patterns (PAMPs), such as lipopolysaccharide and peptidoglycan, are capable of inducing inflammatory responses mediated by the activation of pattern recognition receptors (PRRs), like Toll-like receptor (TLR), in Kupffer cells and hepatic stellate cells, leading to liver injury and fibrosis [19,20,21]. Besides, some metabolic byproducts of bacterial metabolism may interfere with glucose and lipid metabolism, as discussed below, contributing to the exacerbation of liver disease [22,23]. On the other hand, it is well known that GM and bile acids (BAs) closely interact and modulate each other; BAs prevent intestinal bacterial overgrowth and subsequent gut barrier dysfunction, and the GM regulates bile acid composition [24]. Given that BAs modulate host metabolism and immunity, through farnesoid X receptor (FXR) and membrane-associated G protein-coupled receptor (TGR5) signaling, an imbalance in gut bacteria and BAs may trigger metabolic diseases, such as NAFLD [25] (Figure 1).

Specific GM alterations have been correlated with the development and progression of NAFLD, both in human and in experimental animal models [6,22,26,27,28]. NAFLD patients exhibit more Gram-negative and fewer Gram-positive bacteria compared to healthy subjects, and disease progression correlates with phylum-level changes, such as an increase in Proteobacteria and a decrease in Firmicutes [29]. At the genus level, a significant increase in the abundance of Bacteroides and a decrease in Prevotella was observed in NASH patients, when compared to NAFLD patients without NASH [30]. Increased abundance of Ruminococcus in patients with fibrosis was also reported, as well as a relative increase in Streptococcus in obese patients with NAFLD [31]. These alterations are specifically linked to hepatic conditions, and are not the byproduct of insulin resistance, as demonstrated by Da Silva et al. [32]. However, although results point to a correlation between the GM and liver condition, the particular bacterial species involved are largely discordant across individual studies. These inconsistent results may be attributed to the lack of control and regularization for factors known to severely impact the GM, such as weight, diet, and drug intake [33,34]. Additionally, restricting the analysis to changes in diversity indices or comparisons at the phylum and other high-rank taxonomic levels is unlikely to yield insight into the molecular mechanisms involved. For these reasons, we are in need of approaches able to infer causal links, rather than mere statistical associations between specific bacterial species and liver conditions. The different methods for characterization of the GM are shown in Figure 2.

Meta-taxonomical approaches, if merely understood as the analysis of 16S sequences, are probably insufficient to unravel the causal links between GM composition and diseased states. Bacterial species contain pleomorphic genomes, with a conserved genetic core conserved among all members of a species, but also an accessory part that is highly variable among individual clones. The accessory part of the genome is often encoded in plasmids and other mobile genetic elements, thus subject to frequent change [35]. Due to this intrinsic genomic plasticity, strains of the same species frequently display significant phenotypic differences. Alternate metabolic profiles and even distinct virulence levels are common among strains of the same species, as exemplified by pathogenic and commensal *E. coli* [36,37,38]. As a consequence, meta-taxonomy alone may be unable to discriminate between strains that promote hepatic damage from those that do not. Similarly, if hepatic damage is the by-product of bacterial metabolism, it is likely that strains from different species produce similar hepatotoxic compounds as, in many species, non-essential, adaptive metabolic pathways are often encoded in mobile genetic elements [39,40]. 

Although the specific bacterial strains and species involved in NAFLD are still unknown, there is ample evidence that GM perturbations have a causal role in the development of the disease, rather than being a mere consequence of it. In animal models, it was shown that introducing a conventional GM in axenic mice increased monosaccharide absorption and triggered liver lipogenesis [41]. Similarly, faecal transplants from human donors with hepatic steatosis triggered a rapid development of hepatic steatosis in mice [42]. These phenomena suggest that GM does affect the host energy metabolism and fat storage. It may be thus key in the systemic inflammation associated with obesity, which leads to insulin resistance and hepatic steatosis. The molecular mechanisms by which GM alterations translate into hepatic damage are uncertain. However, several studies identified microbial metabolites associated with NAFLD, suggesting a role of certain gut-microbiome-derived metabolites in the pathogenesis and progression of NAFLD [42,43,44,45].

## 3. Microbial Metabolic Pathways

The GM exhibits an array of metabolic pathways that generate multiple final products able to cross the intestinal barrier [46]. Some of these metabolites may provide a benefit for the health of the host via regulation of immunity, supplementation of nutrition and homeostasis [47]. On the other hand, other bacterial metabolites may deregulate intestinal permeability and bile acid metabolism causing liver injury. In this sense, gut microbial unbalances have been shown to be associated with changes in the level of serum metabolites [43].

### 3.1. Microbial Fermentative Pathways

Ethanol is an important microbial metabolite. Several studies reported high alcohol production by the microbiota of some patients with NAFLD [22,48,49]. Gut microbiota have alcohol-metabolizing enzymes such as alcohol dehydrogenase, which converts ethanol into acetaldehyde and acetate [50]. Acetaldehyde has been implicated in weakening the intestinal tight junctions, compromising the gut barrier and enabling translocation of microbial products [51,52]. Besides, the mucosa of the gastrointestinal tract absorbs ethanol by simple diffusion, and the liver responds by upregulating its own ethanol metabolic pathways [50,53]. Ethanol-metabolizing enzymes such as alcohol dehydrogenase, aldehyde dehydrogenase, and catalase are upregulated in NASH liver, strongly suggesting that alcohol metabolism may be an important trigger of NAFLD pathogenesis [54,55]. This was proven in a recent study, where the introduction of strains of Klebsiella pneumoniae with high alcohol production induced NAFLD in mice [56]. Metabolization of ethanol in the liver may contribute to the formation of free fatty acids and oxidative stress, but further studies are required to determine the effects of this metabolite on NAFLD progression. The GM produces other hepatotoxic alcohols through metabolic pathways other than fermentation. Methanol, for example, is produced by certain species as a by-product of pectin metabolization and vitamin B-12 synthesis [57].

Endogenous ethanol production by the gut microbiota may explain the similarities between NASH, alcoholic fatty liver disease [22], and “auto-brewery” syndrome, where subjects suffer from alcohol intoxication after ingesting carbohydrate-rich meals [58]. Although auto-brewery syndrome is a rare, extreme condition, it illustrates how alterations in microbial composition may cause a huge metabolic imbalance. If an abnormal configuration of the normal gut microbiota can even produce an alcohol intoxication, it is conceivable that a less severe situation may produce enough ethanol to inflict chronic damage to the liver.

The GM does not only produce ethanol as a final fermentation product (Figure 3). Diverse members of the GM ferment complex carbohydrates, such as those present in dietary fiber, to produce short-chain fatty acids (SCFAs), such as acetate, propionate and butyrate. Although most SCFAs are consumed in the gut, some are absorbed by the gut epithelium, reaching the liver through the portal vein, where they take part in gluconeogenesis (propionate participates in glucose metabolism) and lipogenesis (acetate and butyrate are potential substrates for lipid synthesis) [59]. Besides, butyrate is an energy source for the enterocytes and facilitates maintenance of the intestinal barrier [60,61]. A reduction in butyrate is linked to weakening of intestinal tight junctions and, hence, permeability [62].

Many functions of SCFAs are mediated by G-protein coupled receptors (GCPRs), which are expressed in intestinal enteroendocrine cells, adipocytes and immune cells. For this reason, SCFAs regulate secretion of gut hormones like glucagon-like peptide-1 (GLP-1) and peptide YY (PYY), inhibit lipolysis and promote adipocyte differentiation, and regulate immune response [63,64,65,66]. However, there are contradictory studies about the relationship between SCFAs and risk of metabolic disorders. An increase in butyrate-producing bacteria prevents diet-induced liver steatosis in murine models [67]. A decrease in butyrate levels was also observed in diabetic patients [68]. Moreover, a randomized clinical trial demonstrated that patients with type 2 diabetes under a high-fiber diet showed significant improvement in hemoglobin A1c (HbA1c) levels. This effect is probably mediated by an increase in acetate- and butyrate-producing gut bacterial strains, accompanied by increased GLP-1 production [69]. On the other hand, higher fecal SCFA concentrations were found in genetically obese mice, and were associated with increased gut permeability, excess adiposity and cardiometabolic risk factors [7,70].

### 3.2. Choline Metabolim

Choline is a quaternary ammonium alcohol, which is an important component of cell membrane phospholipids, and is key to liver fat metabolism. Although choline can be synthesized by humans de novo, its endogenous synthesis is insufficient for health, and has to be complemented by dietary intake (it is, thus, an essential nutrient). In the liver, choline is converted to phosphatidylcholine (lecithin) and other phospholipids, which are essential components of cell membranes. Besides, lecithin assists in the excretion of VLDL particles, preventing hepatic accumulation of triglycerides. For this reason, rodents fed a choline-deficient diet are used as model of NASH [71,72]. 

Choline and L-carnitine (a related quaternary ammonium compound) can also be converted to trimethylamine (TMA) by intestinal bacteria, which is absorbed by intestinal epithelial cells. In the liver, TMA is oxidized by the enzyme flavin mono-oxygenase 3 (FMO3) to generate trimethylamine N-oxide (TMAO) [73]. Hence, gut microbiota unbalances can potentially induce an enhanced conversion of choline to TMA, leading to choline deficiency and contributing to NASH [45,74]. Additionally, metabolomics studies in humans identified TMAO as a predictor of thrombotic events, linked to its contribution to platelet hyperreactivity [75]. This result is also supported by the finding that dietary supplementation of mice with TMAO promoted atherosclerosis [75,76,77]. The TMAO pathway is also linked to the pathogenesis of obesity, since FMO3 regulates white adipose tissue [78]. Because of their involvement in glucose and lipid metabolism [45,79,80], FMO3 and TMAO have been associated with NAFLD. However, in vivo studies have produced divergent results on glucose tolerance. In one study where mice were fed with a high-fat diet, TMAO supplementation reduced glucose tolerance, whereas other studies reported improvements in glucose tolerance after chronic TMAO administration [81,82].

### 3.3. Amino Acid Metabolism

Branched-chain amino acids (BCAA) and aromatic amino acids (AAA) have also been associated with gut metabolic unbalances [42,43]. BCAAs, such as leucine, valine and isoleucine, increase in individuals with insulin resistance, has and have a central role in metabolic disorders [83,84]. A study in mice showed that 3-hydroxyisobutyrate (3-HIB), a catabolic intermediate of the BCAA valine, synthesized in muscle cells, stimulates muscle fatty acid uptake, promoting lipid accumulation and to insulin resistance [85]. Tryptophan, an AAA, can be processed by the GM to produce indole. Indole and its derivatives have been demonstrated to upregulate tight junction protein expression, to downregulate pro-inflammatory cytokines production, and to increase the secretion of GLP-1 in the intestinal epithelium [86,87,88]. Despite these beneficial effects, in the liver, indole can be converted to indoxyl sulfate (IndS), an uremic toxin associated with enhanced endothelial dysfunction and increased oxidative stress [89,90]. Interestingly, a recent study provided evidence of a link between a microbial product of AAA metabolism, 3-(4-hydroxyphenyl)lactate, and hepatic steatosis and fibrosis [44]. However, the functional significance of the latter metabolite is unknown. To highlight the ability of microbial compounds to directly affect the hepatic steatosis phenome, Hoyles et al. treated primary human hepatocyte and mice with phenylacetic acid, an AAA-derived microbial metabolite, which resulted in altered BCAA metabolism and hepatic steatosis [42].

All these studies mentioned above suggest causal roles for diverse microbiota-derived metabolites in the development of NAFLD. However, it is currently impossible to dissect the relative contribution of each of these different molecules. In order to do so, we need well-designed studies in humans that take into account the complex interactions with confounding effects. It is known that the GM of mammals suffers important temporal variations, associated with diet, seasonal changes and drug intake among other factors [91,92,93]. Therefore, if an unbalanced microbiota mediates the progression from NAFLD to NASH, it should persist for a period of time long enough to inflict chronic damage. We believe that a possible way for such abnormal microbiota to persist chronically may reside in the microbial metabolic products per se, especially those produced by fermentative pathways. Fermentation products like ethanol and other alcohols are toxic to most microorganisms, but many fermentative species exhibit relatively high tolerance levels to these toxic compounds. A diet rich in sugars may thus boost a vicious circle, such as that shown in Figure 4. An abundance of easily fermentable saccharides may favor fast fermentative routes over slower, but more energetically efficient pathways. This way, in an energy-rich environment, growth of certain fermentative microorganisms would be increased, which would result in the accumulation of metabolic products, toxic to other microbial species producing different fermentative products (e.g., alcohol-producing organisms vs acid-producing organisms; see Figure 3). These toxic compounds (alcohols) may decrease the ability of acid-fermentative species to thrive, helping alcohol-producing microorganisms to colonize the environment. In this way, a diet abnormally high in mono and di-saccharides may boost chronic colonization by an unbalanced microbiota.

## 4. Microbiome-Based Signature 

Given the multifactorial nature and complexity of NASH pathogenesis, we are in need of tools able to characterize the main pathogenic metabolic pathways that may be involved in NASH progression in individual patients. While there is ample evidence of the causal role played by the GM, and the therapeutic opportunities it represents, as mentioned above, we should not ignore its ample potential as a diagnostic tool. If the GM plays a primary role in NAFLD and NASH, it should be possible to identify microbiome signatures in patients under different pathological conditions. As indicated in previous paragraphs, meta-taxonomy and metagenomics alone are probably insufficient to reveal such signatures, since strains from different species may present similar metabolic pathways, and vice versa (strains of the same species frequently differ in their accessory genomes, which may contain important secondary metabolism pathways). A multi-omic approach is probably required to identify such signatures. Unfortunately, most studies addressing the role of the GM in NAFLD and NASH performed to date relied on a single -omic approach [22,29,42,44,48,49]. One remarkable exception is the study performed by Hoyles et al., who integrated stool metagenomics, whole liver transcriptomics, plasma and urine metabolomics, and clinical data obtained from European non-diabetic obese women. This approach allowed the authors to identify a set of molecular pathways originating from the GM that led to hepatic steatosis [42]. Whether this molecular signature is cohort-specific, or it can accurately mark the presence of hepatic steatosis in other populations, requires further investigation. Hopefully, such molecular signatures may be used to detect patients at risk of developing NASH and cirrhosis. This possibility is supported by studies showing strong statistical correlations between GM signatures and the presence of advanced fibrosis and cirrhosis in NAFLD patients [29,94]. Besides, Caussy C et al. determined a gut microbiome signature of NAFLD-cirrhosis, confirming the result with a validation cohort of first-degree relatives of the patients with NAFLD-cirrhosis [94]. This indicates that this signature is valid even if there is a shared gut-microbiome profile, such as exists among biologically related individuals, that is, among subjects with high risk of advanced fibrosis. Given the limitations of other non-invasive methods for diagnosis of advanced fibrosis, such as transient elastography and laboratory tests in the context of high-risk populations (obese and type 2 diabetic individuals) [95,96], GM signatures acquire great clinical relevance. While further studies are required to verify that these statistical correlations are sustained by causal links, several lines of research point out to the power of microbial biomarkers as innovative, non-invasive diagnostic tools.

One key question that remains to be elucidated is the impact of the natural variability of the GM in disease progression and treatment response. There is some evidence that discrepancies in these parameters may be due, at least in part, to personal differences in GM composition [97]. As an example, differences in the success rate of bariatric surgery are associated with specific GM profiles [98]. Therefore, the identification of genes which have an effect on the production of toxic bacterial metabolites involved in NAFLD progression will provide personalized therapeutic targets and non-invasive biomarkers for NASH diagnosis and severity stratification.

## 5. Gut Microbiome-Targeted Therapy

Among current approaches for treating NAFLD, lifestyle-change-based therapies involving dietary improvement and regular exercise remain the treatment of choice for NAFLD patients. Although these interventions have proven useful [99], a completely effective NASH treatment has yet to be developed. The heterogeneity in the response to treatment likely reflects important individual differences in the many factors that influence NAFLD onset and progress. Personalized approaches tailored to the needs of individuals or stratified groups of patients are required. To this end, the potential of intervening on the gut microbiota represents a promising possibility.

There is ample evidence that it is possible to influence GM composition, and interventions through FMT or pro- and prebiotic administration have been shown to successfully alter the abundance of microbial-derived metabolites in murine models [100,101]. These lines of evidence support that targeting the metabolic pathways that produce harmful and beneficial metabolites may represent a novel strategy to prevent or alleviate NASH (Figure 5). 

Prebiotics and probiotics constitute a simple and affordable route for GM modification. Studies on animal models have provided evidence that prebiotics and probiotics possess modulatory effects on the GM and contribute to the NAFLD treatment by improving the abnormal lipid metabolism and gut dysbiosis [102,103,104]. In humans, the use of prebiotics and/or probiotics is associated with a small decrease in BMI and in serum ALT/AST levels [105]. Unfortunately, none of the prebiotic or probiotic trials performed in human subjects included an evaluation of the liver histological outcome [97]. Another possible strategy is to employ antibiotics. Antibiotic therapy, however, is known to alter a large range of the microbial species, and, while some results indicated that this could have a positive impact on NAFLD progression, no conclusive results have been obtained yet [106,107,108].

FMT represents a straightforward approach, aimed at replacing a pathological GM with that of a healthy individual [109]. This treatment has become common in recent years for patients with relapsing colitis caused by *Clostridium difficile*. In this pathology, FMT achieves high cure rates at a much lower cost than classical antibiotic treatment [110,111]. FMT has been tested in other pathologies, such as inflammatory bowel disease, autism and acute graft-versus-host disease [112,113,114,115]. To date, no FMT studies have reported on the impact of this intervention on NAFLD patients. Experimental studies on mouse models have shown that FMT reduces intestinal permeability, improves intrahepatic lipid accumulation and insulin resistance, and increases pro-inflammatory cytokines [116,117]. FMT was found to restore GM diversity, increasing Bacterioidetes and reducing Actinobacteria and Firmicutes, with a concomitant increase in butyrate production, a SCFA with local anti-inflammatory effects [118]. Recent data showed that FMT from human donors with NAFLD triggered steatosis in recipient mice [42]. In humans, a randomized controlled trial where patients with metabolic syndrome received the gut microbiota from lean, healthy subjects, showed an increase insulin sensitivity after 6 weeks of FMT [119]. These studies suggest that FMT may constitute a suitable approach to restoring a “physiological” intestinal microbiome, highlighting the current need to perform clinical trials addressing FMT in NAFLD [109].

An issue to be resolved with regard to FMT is the stability of microbiota post-FMT. A problem found in several studies regarding the use of FMT for the treatment of chronic diseases is the loss of response over time. This is probably due to the lack of FMT persistence, and the reversion of the GM to its previous, pathological composition. In fact, it has been shown that in chronic conditions such as ulcerative colitis, repeated FMT treatments are superior to single administration procedures [120]. Regarding the persistence of the FMT, a study of 14 patients with recurrent *Clostridium difficile* infection noted that engraftment (the percentage of the community in patient samples that is attributable to donor communities) persisted through a 1-year follow-up [121]. The length of the FMT engraftment, however, is likely to depend on the administration route and the pathological condition considered. A recent study evaluated bacterial engraftment following encapsulated delivery of fecal microbiota, measuring donor bacterial persistence in 18 patients with a positive FMT response, that did not experience post-transplant *C. difficile* recurrences nor required further antibiotic exposure [122]. A majority of the patients (61%) demonstrated that donor engraftment persisted even after one year. However, the authors also noted substantial variability in long-term bacterial engraftment, regardless of clinical outcome. This indicates that neither complete nor sustained donor engraftment is necessary for long-term clinical recovery, and the FMT may simply act as a disruption strong enough to drive the reorganization of the microbial community into a healthy state. Besides, recent studies have revealed the existence of distinct and niche-specific microbial communities along the gastrointestinal tract influencing engraftment rates of exogenous microbes, suggesting that future microbial interventions may be personalized based on an individual’s gut microbiome [123,124]. Further mechanistic investigations are necessary to characterize functional changes and potential functional redundancies associated with the various efficacious reorganizations of the intestinal microbiota.

Another aspect related to FMT that requires further analysis is the strategy involved in donor selection. Currently, there are strict recommendations for selection and screening of fecal donors to prevent disease transmission [125,126]. But other aspects apart from the transmission of pathogens require consideration. For example, recent studies have shown the onset of obesity in patients receiving stool from a healthy but overweight donor [127]. This indicates that other aspects such as a normal BMI may require consideration before selecting a suitable FMT donor.

If antibiotic treatment and FMT represent somewhat radical procedures to intervene in the GM, current research focuses on more detailed, specific ways to engineer the microbiome. The goal of these approaches, still in preliminary stages, is to specifically target bacteria from certain species or carrying certain genes. One way to achieve this is the use of bacteriophages, bacterial viruses that are highly specific to certain strains/molecular receptors. These agents can be used to attack specifically the pathogenic components of the GM and represent a promising line of research in the context of ever-rising antibiotic resistance [128]. In humanized mice colonized with bacteria from the faeces of patients with alcoholic hepatitis, Duan et al. investigated the therapeutic effects of bacteriophages that targeted *Enterococcus faecalis*. This species was enriched in the faecal samples from alcoholic hepatitis patients, and targeting it with bacteriophages alleviated ethanol-induced liver injury [129]. The exquisite specificity of bacteriophages made them popular tools for antimicrobial treatment, yet their applicability is hampered by the frequent emergence of resistances. Current approaches to microbiome engineering make use of bacteriophages in combination with other molecular tools such as CRISPR-Cas systems to target not only a particular species, but cells that contain a specific gene [130]. While still in their infancy, such approaches may allow us to eliminate the bacterial strains responsible for the production of toxic metabolites with surgical precision.

## 6. Conclusions

It is now widely accepted that changes to the composition and function of the GM are associated with NAFLD. However, a comprehensive understanding of the interactions between the microbiome and the liver still evades us. Many of the effects seem mediated by metabolites produced by commensal bacteria utilizing dietary nutrients as precursors, which does not seem to depend on particular species but rather on microbial metabolic pathways.

Integrated multi-omic analyses of GM allowed the identification of possible microbial-driven mechanistic pathways in NAFLD and provided preliminary evidence that it is possible to use a microbiome-derived metagenomics signature to detect advanced NASH. These approaches have led to the identification of candidates for microbiome-targeted therapy. Nevertheless, large-scale multi-omic studies are still required to determine the effects of endogenous production of the microbiota-derived metabolites on NAFLD progression. The safety and efficiency of novel therapeutic strategies to engineer the GM also need to be thoroughly analyzed. With limited treatments currently available against NASH, microbiota-targeted interventions deserve further exploration as a potential therapeutic option against NAFLD progression.

## Figures and Tables

**Figure 1 jcm-09-01369-f001:**
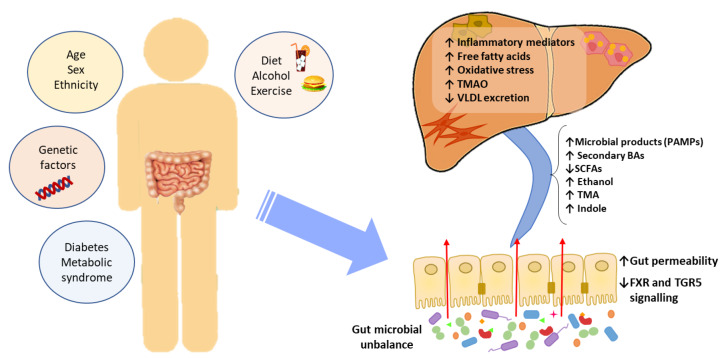
The effect of gut microbial unbalance in NAFLD. Different factors affect the gut microbiome. Gut microbial unbalance causes an increase in secondary BAs, which modulates FXR and FGR5 signaling, affecting the glucose and lipid metabolism and anti-inflammatory immune response. Besides, the increase in certain microbial metabolites mediates weakening of intestinal tight junction, enabling passage to the systemic circulation of PAMPs and microbial metabolites (such as ethanol) that reach the liver inducing inflammatory responses, liver injury and fibrosis. BAs, bile acids; FXR, farnesoid X receptor; PAMPs, pathogen associated molecular patterns; SCFAs, short-chain fatty acids; TGR5, membrane-associated G protein-coupled receptor; TMA, trimethylamine; TMAO, trimethylamine N-oxide; VLDL, very low-density lipoprotein.

**Figure 2 jcm-09-01369-f002:**
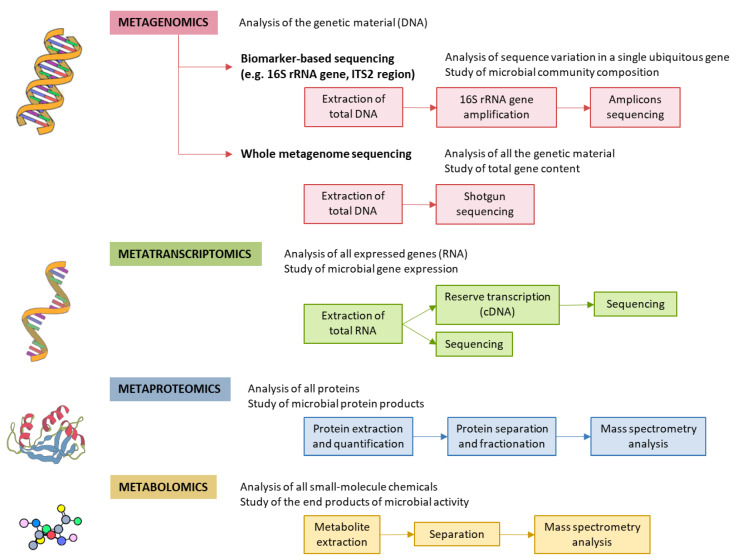
Methods for characterizing gut microbiota. 16S rRNA is highly conserved among bacterial species, except that it contains hypervariable regions that confer phylogenetic association; thus, 16S rRNA gene sequencing is widely used for phylogenetic reconstruction and quantification of microbial diversity. However, this technique does not make it possible to decipher functional changes in the microbiome or to find out the true impact of gut microbes on disease states. For this reason, several -omics approaches were put forward. These methods dig into genes for genetic information storage, transcription for gene expression, proteins for structural and metabolic activities, and metabolites for end products of metabolism. cDNA, complementary DNA; ITS2, internal transcribed spacer 2; rRNA, ribosomal RNA.

**Figure 3 jcm-09-01369-f003:**
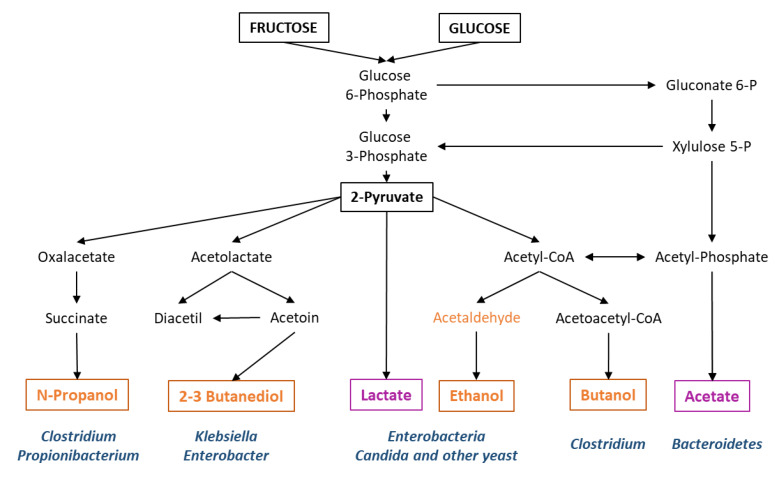
Major microbial fermentative pathways and bacterial genera frequently involved. In orange, those final metabolites for which there is evidence of liver damage. In purple, metabolites for which a protective effect is suggested.

**Figure 4 jcm-09-01369-f004:**
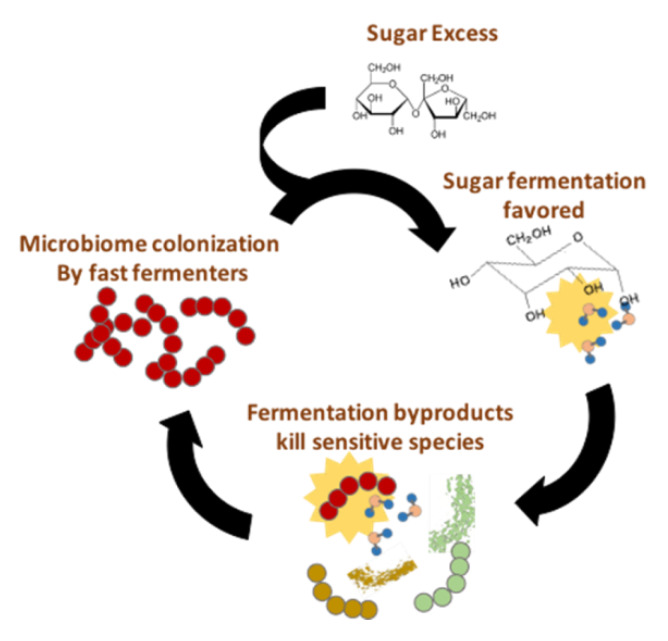
A sugar-rich diet may boost a vicious circle sustaining an alcohol-producing fermentative microbiota.

**Figure 5 jcm-09-01369-f005:**
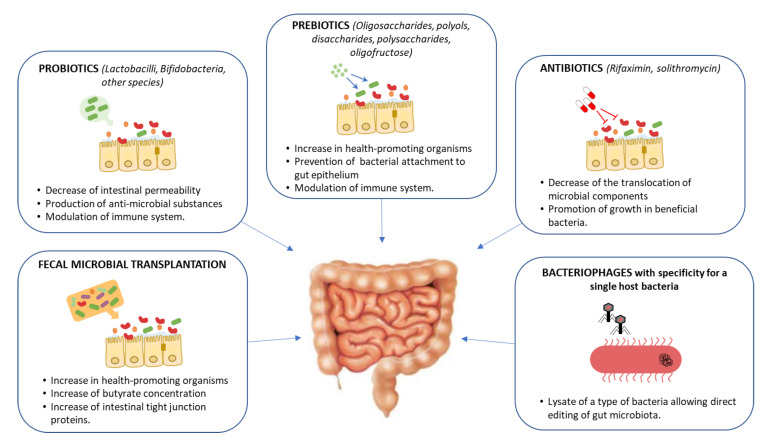
Potential gut-microbiome-targeted therapies in hepatic disease and its mechanisms of action.

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
