# Peer review of "A Role for Gut Microbiome Fermentative Pathways in Fatty Liver Disease Progression"

_jcm, 2020, doi:10.3390/jcm9051369_

Round 1

Reviewer 1 Report

This is a well written review that summarizes the evidence regarding the association of NAFLD with gut microbiome. It gives adequate information for the reader on several aspects of this relationship.

My only concern is that section #5 (Gut microbiome-targeted therapy) seems somehow unilateral emphasizing on the role of fecal transplantation. Data on therapy are definitely scarce and should be presented in a balanced way to avoid possible bias. 

Author Response

Many thanks for your comments. We have emphasized on the role of fecal transplantation due to the growing interest it is having in the field of metabolic diseases, but, in addition, we have special interest on it since we are going to start a clinical trial in NASH with this therapy. Even son, it is true that there is an imbalance in this section, so we have shortened this part.

Reviewer 2 Report

This comprehensive review is well written and of relevance for the scientific community working on liver diseases. A few comments should however be considered.

(1) The title is too general and does not fully represent the content of the review.

(2) The roles of microbiota-derived LPS and peptidoglycans in NAFLD development (e.g. TLR2 / TLR4 signaling) have not been discussed, but would make the review more complete. PAMPs are known as a key components of ‘microbiome-induced NAFLD/NASH’ and could be addressed more comprehensively.

(3) The authors mainly focus on ethanol as a factor that alters microbiome composition in the gut. Yet, bile acids are known to ‘shape’ the gut microbiome. This could also be improved in the manuscript. (This is also of relevance since, some of anti-NASH compounds under development eg targeting FXR might have a consequence  for the gut microbiome. Endogenous bile acids can also agonize FXR influencing bile acid generation and consequently also modifying the gut microbiota.)

Author Response

First of all, we would like to thank you for your kind comments and suggestions for improving the review.

With this review we wanted to highlight the role of microbial metabolites in the pathogenesis of NASH, and we had hardly mentioned other mediators of liver damage related to the gut microbiota, PAMPs and bile acid imbalance. We hope that the changes made to the text are of your pleasure.  

Reviewer 3 Report

I read the article titled “Fatty Liver Disease as a Microbial Disease” by Dr. Iruzubieta and colleagues with great interest.  The topic is  novel and the data in this field is limited.  The review article tries to summarize the literature in this field, However there are multiple flaws in the manuscript.  The manuscript overemphasis the role of gut microbiome and frequently cites multiple review articles to support the statements. Please see below for my specific comments.

  1. The authors have consistently cited review articles as references. This is not appropriate, I encourage to authors to thoroughly revise the manuscript so the statements put forward in the manuscript are supported by clinical trials.

  1. Authors have made multiple strong statements without any references please review the following:

  • Line 40-42: Please add reference to support the statement.

  • Line 42: Authors make claim that gut microbiome is a key player in pathogenesis of Nash. They cite one review are article to support this. Please cite randomized trials to support this statement.

  • Line 54-55: Please cite clinical trial to support this statement.

  • Line 60-61 needs a reference

  • Line 191-192 needs a reference

  • Line 203-204 needs a reference

  1. The authors frequently loose the focus. For example they had started describing methods for FMT (lines 321-325), or the measures COVID-19 pandemic  (lines 355-358)

Author Response

Thanks for your comments, we agree with them and we have proceeded to make the changes. We hope these are of your pleasure.

Reviewer 4 Report

This is a very good review which discusses all aspects of the interaction between microbiom and liver. I strongly agree with the authors "that metataxonomical approaches are probably insufficient to unravel the causal links". In my opinion, the review contributes to a better understanding of the liver gut axis.

Author Response

Many thanks for your kind comment. We have made some minor changes to the text following suggestions from the reviewers. We hope the review continues to be of your pleasure. 

Round 2

Reviewer 3 Report

It appears that the authors may have revised the manuscript. I'm unable to track what changes were made. It is expected the authors provided point by point response to each comment made by reviewer. 

Author Response

Sorry about that.

Below you can find the answers point by point to your comments and we attach the file with the corrections.

Line 40-42: Please add reference to support the statement.

DOI: 10.1007/s11901-017-0378-2

Line 42: Authors make claim that gut microbiome is a key player in pathogenesis of Nash. They cite one review are article to support this. Please cite randomized trials to support this statement.

DOI: 10.1152/physrev.2001.81.3.1031

DOI: 10.1053/j.gastro.2010.11.049

DOI: 10.1038/nature05414

Line 54-55: Please cite clinical trial to support this statement.

DOI: 10.1073/pnas.0808567105

DOI: 10.1073/pnas.0601056103

DOI: 10.1038/nature07540

DOI: 10.1038/nature05414

Line 60-61 needs a reference

DOI: 10.1016/s0168-8278(00)80242-1

DOI: 10.1155/2010/453563

DOI: 10.1016/j.cmet.2016.05.005

Line 191-192 needs a reference

DOI: 10.1038/srep19076

DOI: 10.2337/db06-0097

Line 203-204 needs a reference

DOI: 10.1038/s41591-018-0061-3

DOI: 10.1038/nature18646

The authors frequently loose the focus. For example they had started describing methods for FMT (lines 321-325), or the measures COVID-19 pandemic  (lines 355-358)

We have deleted those paragraphs.

Round 3

Reviewer 3 Report

Comments have been addressed satisfactorily by the authors.